# Evolutionary analyses of the major variant surface antigen-encoding genes reveal population structure of *Plasmodium falciparum* within and between continents

Gerry Tonkin-Hill[1,2,3], Shazia Ruybal-Pesántez[1¤], Kathryn E. Tiedje[1,4], Virginie Rougeron[5], Michael F. Duffy[1,4], Sedigheh Zakeri[6], Tepanata Pumpaibool[7,8], Pongchai Harnyuttanakorn[8,9], OraLee H. Branch[10,11], Lastenia Ruiz-Mesía[11], Thomas S. Rask[1], Franck Prugnolle[5], Anthony T. Papenfuss[2,12,13,14,15], Yao-ban Chan[12,16], Karen P. Day[1,4]*

1 School of BioSciences, Bio21 Institute, The University of Melbourne, Melbourne, Australia,
2 Bioinformatics Division, Walter and Eliza Hall Institute, Melbourne, Australia, 3 Parasites and Microbes, Wellcome Sanger Institute, Wellcome Genome Campus, Hinxton, United Kingdom, 4 Department of Microbiology and Immunology, Bio21 Institute and Peter Doherty Institute, The University of Melbourne, Melbourne, Australia, 5 Laboratoire MIVEGEC, Université de Montpellier-CNRS-IRD, Montpellier, France, 6 Malaria and Vector Research Group (MVRG), Biotechnology Research Center, Pasteur Institute of Iran, Tehran, Iran, 7 Biomedical Science, Graduate School, Chulalongkorn University, Bangkok, Thailand, 8 Malaria Research Programme, College of Public Health Science, Chulalongkorn University, Bangkok, Thailand, 9 Department of Biology, Faculty of Science, Chulalongkorn University, Bangkok, Thailand, 10 Concordia University, Portland, Oregon, United States of America, 11 Universidad Nacional de la Amazonía Peruana, Iquitos, Perú, 12 School of Mathematics and Statistics, The University of Melbourne, Melbourne, Australia, 13 Peter MacCallum Cancer Centre, Victorian Comprehensive Cancer Centre, Melbourne, Australia, 14 Department of Medical Biology, The University of Melbourne, Melbourne, Australia, 15 Sir Peter MacCallum Department of Oncology, The University of Melbourne, Melbourne, Australia, 16 Melbourne Integrative Genomics, The University of Melbourne, Melbourne, Australia

¤ Current address: Population Health and Immunity Division, Walter and Eliza Hall Institute, Melbourne, Australia; Department of Medical Biology and Bio21 Institute, The University of Melbourne, Melbourne, Australia; Burnet Institute, Melbourne, Australia
* Karen.Day@unimelb.edu.au

**Data Availability Statement:** The sequences for this project have been deposited at DDBJ/ENA/GenBank: PRJNA385207, PRJNA385208,

## Abstract

Malaria remains a major public health problem in many countries. Unlike influenza and HIV, where diversity in immunodominant surface antigens is understood geographically to inform disease surveillance, relatively little is known about the global population structure of PfEMP1, the major variant surface antigen of the malaria parasite *Plasmodium falciparum*. The complexity of the *var* multigene family that encodes PfEMP1 and that diversifies by recombination, has so far precluded its use in malaria surveillance. Recent studies have demonstrated that cost-effective deep sequencing of the region of *var* genes encoding the PfEMP1 DBLα domain and subsequent classification of within host sequences at 96% identity to define unique DBLα types, can reveal structure and strain dynamics within countries. However, to date there has not been a comprehensive comparison of these DBLα types between countries. By leveraging a bioinformatic approach (jumping hidden Markov model) designed specifically for the analysis of recombination within *var* genes and applying it to a dataset of DBLα types from 10 countries, we are able to describe population structure of

PRJNA630836, KY328840–KY341897, KX845707–KX851405, KP219986- KP221189.The relevant data used to produce the figures are deposited in https://github.com/gtonkinhill/global_var_manuscript.

**Funding:** This research was supported by the National Institute of Allergy and Infectious Disease, National Institutes of Health [Grant number: R01-AI084156 to K.P.D.] (https://www.niaid.nih.gov), Fogarty International Center at the National Institutes of Health [Program on the Ecology and Evolution of Infectious Diseases (EEID), Grant number: R01-TW009670 to K.P.D.] (https://www.fic.nih.gov), and the National Institute of Allergy and Infectious Disease, National Institutes of Health [Grant number: R01-AI149779 to K.P.D.] (https://www.niaid.nih.gov). Salary support was provided by R01-AI084156 to G.T-H, K.E.T, V.R, and T.S.R; R01-TW009670 to K.E.T; The University of Melbourne to K.E.T and M.F.D. S.R-P was supported by a Melbourne International Engagement Award from The University of Melbourne. The funders had no role in study design, data collection and analysis, decision to publish, or preparation of the manuscript.

**Competing interests:** The authors have declared that no competing interests exist.

DBLα types at the global scale. The sensitivity of the approach allows for the comparison of the global dataset to ape samples of *Plasmodium Laverania* species. Our analyses show that the evolution of the parasite population emerging out of Africa underlies current patterns of DBLα type diversity. Most importantly, we can distinguish geographic population structure within Africa between Gabon and Ghana in West Africa and Uganda in East Africa. Our evolutionary findings have translational implications in the context of globalization. Firstly, DBLα type diversity can provide a simple diagnostic framework for geographic surveillance of the rapidly evolving transmission dynamics of *P. falciparum*. It can also inform efforts to understand the presence or absence of global, regional and local population immunity to major surface antigen variants. Additionally, we identify a number of highly conserved DBLα types that are present globally that may be of biological significance and warrant further characterization.

## Author summary

Globalization has led to the spread of pathogens through increased human movement. Microbiologists track epidemics of these pathogens by cataloguing geographic diversity in the genes that encode for variant surface antigens (VSA). Here, we developed a computational approach to explore the evolution of specific DNA sequences of the major VSA gene of the human malaria parasite, *Plasmodium falciparum*. First, we tested the method by comparing DNA sequences of these genes from *P. falciparum* to those of *Plasmodium* species that infect chimpanzees and gorillas. We showed that it could distinguish DNA signatures specific to each species. Next, we asked whether our method could detect geographic signatures within these genes by analyzing a global collection of *P. falciparum* isolates from 23 locations in 10 countries. The important outcome of our work was the ability to identify geographic signatures specific to countries and continents that were consistent with the "out of Africa" origin of *P. falciparum*. We can now identify malaria parasites from countries within Africa, South America, and Asia/Oceania using a diverse region of VSA genes without having to sequence and assemble whole parasite genomes. This methodology has potential applications in malaria surveillance to track parasites as they move around the world.

## Introduction

*Plasmodium falciparum* continues to present a significant economic and public health burden globally. The pathogen is endemic across many resource poor countries such as those in tropical Africa and parts of Asia [1] and re-emerging in Latin America [2]. Part of the pathogen's success in remaining endemic, while also highly prevalent in many regions, can be attributed to the extreme diversity of the major variant surface antigen of the blood stages, known as *P. falciparum* erythrocyte membrane protein 1 (PfEMP1). This molecule is encoded by the *var* multigene family [3] and each parasite possesses approximately 60 *var* genes [4,5]. Genome sequencing has shown that each parasite carries a different *var* gene repertoire [6]. Analysis of the population structure of the *var* genes encoding PfEMP1 is thus important for the control and prevention of the disease as well as for the design of any vaccine targeting PfEMP1 based on an understanding of variant-specific immunity [7].

*P. falciparum* is able to chronically infect humans, in part by evading the host immune system through switching between monoallelic expression of different PfEMP1 isoforms during infection [8–10]. PfEMP1 is expressed during both the trophozoite blood stage and the very early gametocyte transmission stage of the *P. falciparum* life cycle [11,12]. Expression of PfEMP1 on the surface of the infected erythrocyte allows it to adhere to a diverse set of host receptors, helping to evade host defense systems. This leads to pathogenic sequestration of infected erythrocytes in the microvasculature [13]. A consistent finding from case-control studies, including a recent transcriptome analysis, has been that a conserved set of expressed PfEMP1 sequences are associated with severe disease [14–21].

*Var* genes are comprised of multiple alternating semi-conserved Duffy binding-like (DBL) domains and cysteine-rich interdomain regions (CIDRs) [22], which have been further classified into subtypes DBLα, β, γ, δ, ε, ζ, x and CIDRα, β, γ, and δ [6,23]. Minor subtypes have also been distinguished, e.g., DBLα0, 1, 2, as well as conserved homology blocks and segments related to severe disease [6,14,24]. *Var* genes have been shown to diversify by meiotic recombination during the obligatory sexual phase of the life cycle [25]. In addition, *in vitro* studies have pointed to mitotic recombination generating sequence diversity during asexual replication [26–28]. In fact, a single break within a *var* gene region has been shown to lead to a cascade of recombination with the generation of multiple chimeric *var* genes *in vitro* [29].

The global population structure of *P. falciparum* has been observed previously using microsatellite loci [30] and subsequently single nucleotide polymorphisms (SNPs) and whole genome sequencing [31–35]. Recent work has used whole genome sequencing to investigate a global dataset of *var* gene sequences but the high cost of such an approach makes it currently impractical for routine surveillance in malaria endemic countries [5]. Nearly all *var* genes encode a single DBLα domain, making it a suitable marker for characterizing population structure. Not only is the DBLα domain highly prevalent, it is also very variable, with the average pairwise identity of the amino acid sequences encoding this domain being approximately 42% [6]. For the purposes of this study, we were interested to determine if the 450bp *var* sequence encoding the DBLα domain could determine geographic population structure.

Previous studies of the DBLα diversity have so far been restricted to specific geographic regions or small sample cohorts. In these studies, we have designated sequences encoding DBLα with less than 96% sequence identity as unique DBLα types. When dealing with field isolates often containing multi-genome infections, DBLα alleles of each member of the *var* multigene family cannot be assigned due to the absence of known chromosomal positions and inability to assign to a specific genome. Given these constraints, DBLα types of an isolate rather than alleles are used for population genetic analyses in regional and local datasets [36]. Barry et al. (2007) [7] investigated the DBLα type diversity found in 89 global isolates from both field and laboratory clones, including 30 isolates from Amele, Papua New Guinea (PNG). They found an extreme diversity of DBLα types at the global level, with a higher level of conservation in the PNG population compared to the global isolates. Due to the limited size of their global dataset, it was not possible to investigate the DBLα type population structure at a global level. Chen et al. (2011) [37] analyzed 160 field isolates from a global collection, confirming the higher DBLα type diversity observed in Africa, and observed that Bakoumba, Gabon appeared to have a higher conservation than the other African countries sampled. Tessema et al. (2015) [38] further investigated the DBLα type population structure in PNG, identifying fine-scale population structure of the DBLα type repertoires at the village level. Rougeron et al. (2017) [39] analyzed a collection of isolates from South America, observing a smaller population size of DBLα types than has been reported in other regions, and suggested that the DBLα type population structure mirrored that found in the SNP and microsatellite analysis of Yalcindag et al. (2012) [31]. Ruybal-Pesántez et al. (2017) [40] investigated DBLα

type diversity across six sites in Uganda, identifying high diversity and little repertoire overlap. Day et al. (2017) [41] reported a similar finding in Gabon, suggesting that the lack of overlap between repertoires was the non-random result of immune selection creating strain structure. Subsequent network analyses and stochastic simulations that consider both epidemiological and evolutionary processes confirmed that frequency-dependent immune selection can structure DBLα type repertoires [42]. Of note, these molecular epidemiological studies showed high diversity of DBLα types and repertoires, with individual DBLα types conserved in space and time within and between sampling sites [36,39–41] despite *in vitro* predictions of rapid evolution by mitotic recombination [27,29].

Here, by leveraging an approach we designed specifically for the analysis of recombination among *var* genes [43] and applying it to a global dataset of DBLα types from 23 locations in 10 countries, we describe DBLα population structure at a global scale. The sensitivity of this approach also allows for a comparison of DBLα sequences isolated from ape samples of other *Plasmodium* species to a global dataset of translated *P. falciparum* DBLα types. Indeed, we identify strong DBLα population structure both globally and within Africa. This contrasts with previous studies, which have struggled to distinguish population structure within Africa using entire *var* genes [5]. The population structure we identified was then related to distant ape species as well as previous population studies of *P. falciparum*. Additionally, we describe a number of globally conserved DBLα types that have not previously been well characterized and may be of high biological significance. The relevance of these findings to contemporary malaria surveillance is discussed.

## Results

### Jumping hidden Markov model

To extend previous approaches that focused on comparing DBLα types between isolates at 96% pairwise sequence identity [7], we adapted the approach of Zilversmit et al. (2013) [43], which reconstructs the translated sequence of each DBLα type in the dataset as an imperfect mosaic of donor sequences using a jumping hidden Markov model (JHMM). For the purposes of this study, this improves upon prior approaches that ignore recombination, as previously, *var* repertoires that share a significant amount of homology could be mis-classified as very distant due to the presence of a recombination event (Fig 1A). We used the JHMM to infer the posterior probability that each location in an isolate's DBLα type amino acid sequence is most closely related to every other DBLα type in our dataset. We then accumulated these probabilities over all DBLα types found in an isolate to provide an estimate of the expected proportion of relatedness between isolates (Fig 1B). These proportions were then aggregated, accounting for repertoire size, to provide estimates of an isolate's DBLα repertoire that most closely matched each donor population (country). This provides a measure of relatedness between each isolate and country (see Materials and Methods for a thorough description of the JHMM). A flowchart outlining the overall analysis pipeline is given in Fig 1C.

### Investigating the *Plasmodium Laverania* genus using the JHMM

Using the JHMM, we first investigated the relationship between DBLα types from the *P. falciparum* laboratory strains 3D7, Dd2 and HB3 with previously-published types from the *P. billcollinsi*, *P. reichenowi*, and *P. praefalciparum* species, which have been found in the great apes [5,45]. This helped to validate our approach and gave insights into the relationship among the DBLα types of different primate species.

Fig 1D illustrates the resulting relatedness after comparing all the *Plasmodium* species to the three laboratory strains, which we used as representatives for *P. falciparum*. As expected,

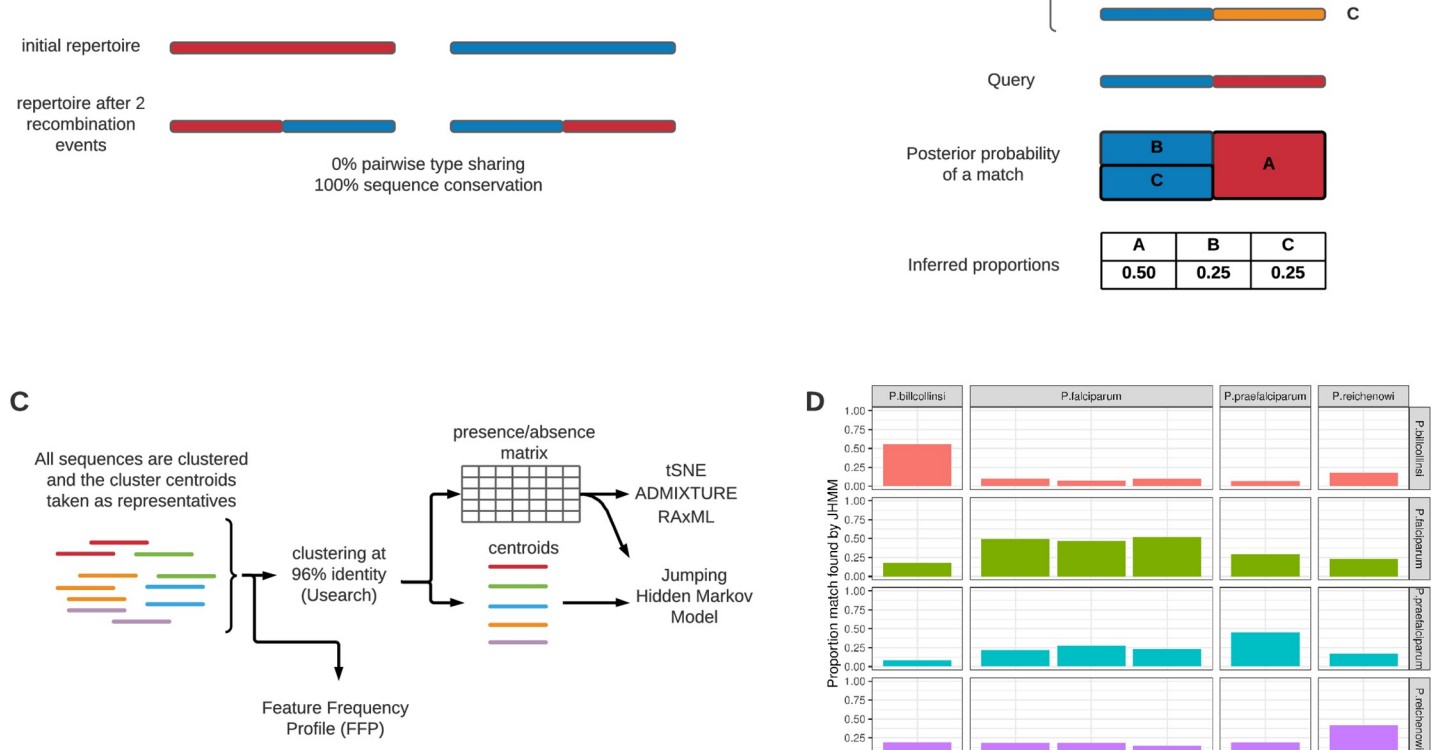

**Fig 1. (A)** The diagram illustrates the issues with only considering DBLα types at the 96% pairwise sequence identity threshold. After just two recombination events (within the same repertoire) this model illustrates that the pairwise type sharing (PTS) will indicate no relatedness (i.e., no DBLα type sharing or 0% PTS) between the initial *var* repertoire and the *var* repertoire after recombination, while the overall sequence composition remains the same (i.e., 100% sequence conservation). Note: This could also occur through a single recombination event involving non-homologous translocation with reciprocal exchange. **(B)** An illustration of the JHMM approach. A query is searched against a database of DBLα types using the JHMM. The resulting posterior probabilities of a match are aggregated into proportions indicating ancestry relationships. **(C)** A simplified flowchart outlining the analysis pipeline. **(D)** The inter-species matching proportions where each column represents an isolate and sums to one. Three *P. falciparum* lines are included in the analysis, 3D7, Dd2, and HB3 (left to right). The *P. praefalciparum* isolate is most closely related to *P. falciparum* before *P. reichenowi* and *P. billcollinsi* in turn, which is consistent with the phylogenetic tree of Larremore et al. (2015) [44]. *P. billcollinsi* has the highest proportion of self-matching, suggesting it is more diverged from the other isolates.

within-species DBLα type comparisons showed the highest matching proportions compared to between-species comparisons, indicating that every sequence is most closely related to other sequences from the same species. When comparing *P. falciparum* to the three ape *Plasmodium* species investigated, the *P. falciparum* DBLα types were most closely related to those of *P. prae-falciparum*. The next closest species was *P. reichenowi*, while *P. billcollinsi* was the most distant. This is consistent with a published phylogenetic tree built from mitochondrial sequences [44] and confirmed genome sequencing of the *Laverania* species [45]. Therefore, the JHMM approach shows that: (a) the population structure of DBLα types is representative of the structure of *Laverania* species as a whole; and (b) this approach is sensitive enough to reconstruct evolutionary relationships even among very diverse sequences. It is important to note that because the inter-species sequence diversity is high, this comparison would not have been possible using the commonly-used pairwise sequence identity threshold of 96%, as none of the DBLα types matched between *P. reichenowi*, *P. billcollinsi*, and *P. falciparum* within the 96% threshold.

**Table 1. Summary information of the global *P. falciparum* isolates included.**

| Country of origin (Population) | Number of isolates | Dates of collection | Malaria disease status | Ages (years) | References |
|---|---|---|---|---|---|
| Uganda (Apac) | 77 | 2006–2007 | Uncomplicated | 1–5 | [40] |
| Uganda (Arua) | 97 | 2006–2007 | Uncomplicated | 1–5 | [40] |
| Uganda (Jinja) | 90 | 2006–2007 | Uncomplicated | 1–5 | [40] |
| Uganda (Kanungu) | 79 | 2006–2007 | Uncomplicated | 1–5 | [40] |
| Uganda (Kyenjojo) | 83 | 2006–2007 | Uncomplicated | 1–5 | [40] |
| Uganda (Tororo) | 91 | 2006–2007 | Uncomplicated | 1–5 | [40] |
| Ghana (Soe) | 108 | 2012 | Asymptomatic | All | [36] |
| Ghana (Vea/Gowrie) | 122 | 2012 | Asymptomatic | All | [36] |
| Gabon (Bakoumba) | 201 | 2000 | Asymptomatic | 1–12 | [41] |
| Peru (Zungarococha/Mazan) * | 13 | 2011 | Asymptomatic | All | [46] |
| Peru (Iquitos) | 21 | 2003–2004 | Uncomplicated | All | [31,39] |
| French Guiana (Camopi) | 41 | 2006–2008 | Uncomplicated | All | [31,39] |
| French Guiana (Trois Sauts) | 35 | 2006–2008 | Uncomplicated | All | [31,39] |
| Venezuela (El Caura) | 10 | 2003–2007 | Uncomplicated | All | [31,39] |
| Colombia (Turbo) | 21 | 2002–2004 | Uncomplicated | All | [31,39] |
| Thailand (Kanchanaburi)* | 8 | 2006 | Uncomplicated | All | [31,47] |
| Thailand (Maehongson)* | 9 | 2005 | Uncomplicated | All | [31,47] |
| Thailand (Ranong)* | 9 | 2006 | Uncomplicated | All | [31,47] |
| Thailand (Tak)* | 8 | 2007 | Uncomplicated | All | [31,47] |
| Thailand (Yala)* | 12 | 2007 | Uncomplicated | All | [31,47] |
| Iran (Sistan-Baluchestan)* | 45 | 2000–2003 | Uncomplicated | All | [31] |
| Papua New Guinea (Mugil)** | 35 | 2006 | Asymptomatic | All | [38] |
| Papua New Guinea (Wosera)** | 33 | 2005 | Asymptomatic | All | [38] |
| Total | 1,248 | - | - | - | - |

*P. falciparum* isolates previously collected from these locations were sequenced in the present study (see Materials and Methods and the references provided for all study details).

**P. falciparum* isolates from these locations were sequenced in Tessema et al. (2015) [38].

## Global population structure of DBLα types

The global dataset of the DBLα types used in this study were sequenced from 1,248 *P. falciparum* isolates (obtained from individuals with asymptomatic infections or individuals presenting with uncomplicated malaria at the time of sample collection, see Table 1) across 23 locations in Colombia, French Guiana, Gabon, Ghana, Iran, Papua New Guinea (PNG), Peru, Thailand, Uganda, and Venezuela (i.e., 10 countries worldwide, Fig 2A) [31,36,38–41,46,47]. Clustering of all the DBLα types at 96% identity resulted in 32,682 unique DBLα types used for subsequent analyses. The median number of DBLα types per isolate varied significantly between countries (range = 19–75), with African countries having the highest median and maximum number of types, consistent with a higher multiplicity of infection, that is a higher number of isolates having more than one distinct *P. falciparum* genome (S1 Fig). PNG reported the lowest median number of types, which is likely due to a less sensitive experimental protocol using Sanger sequencing for the PNG isolates [38].

Geographic population structure of DBLα types stratified by country of origin (Fig 2A) was assessed using a binary presence/absence matrix with results shown as a t-Distributed Stochastic Neighbor Embedding (t-SNE) plot (Fig 2B) [48]. Our results show a clear division by continent with the African populations, Asian/Oceanian populations and South American populations forming distinct clusters. Country specific clustering was also observed even

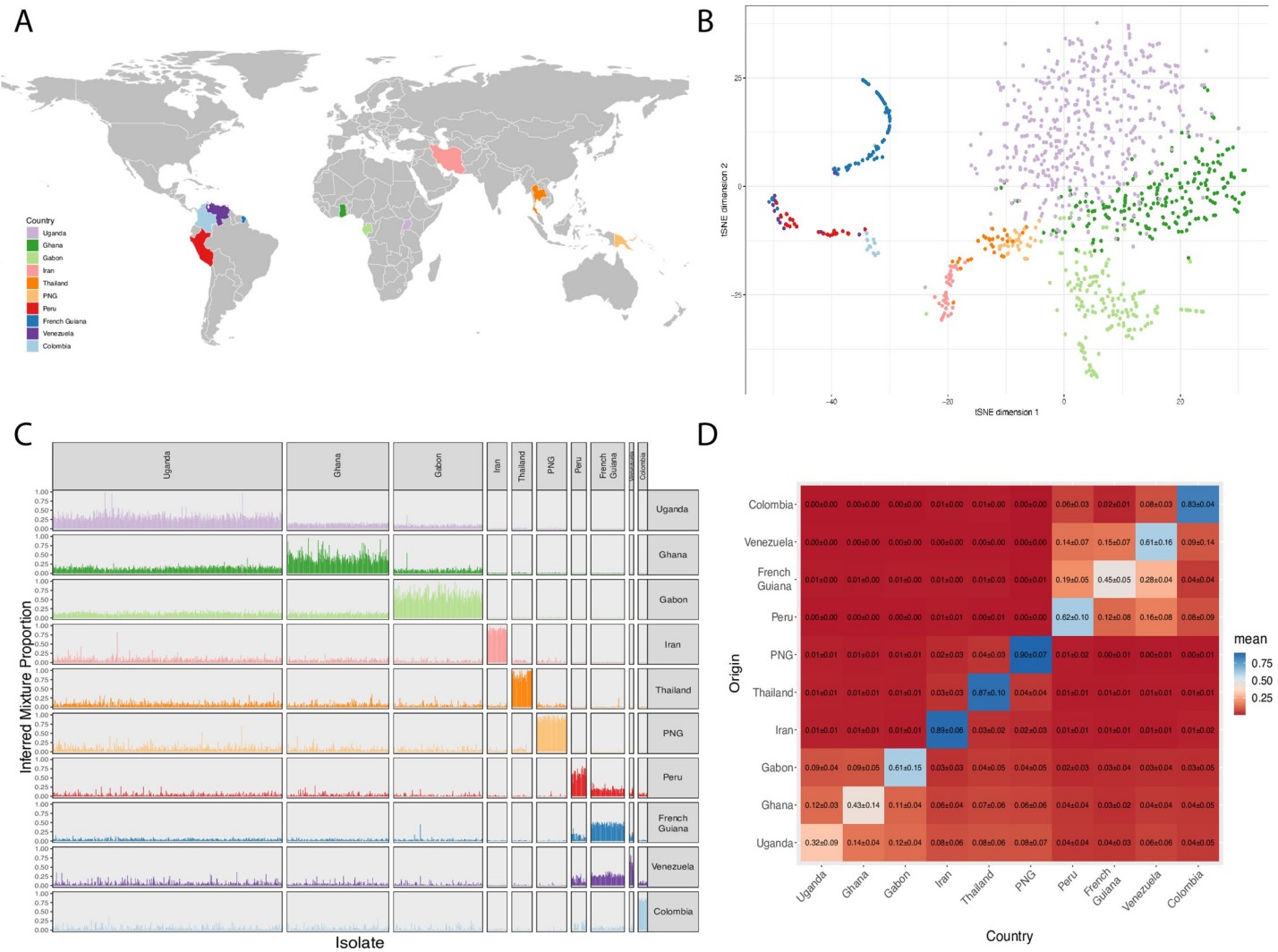

**Fig 2.** **(A)** A world map indicating the countries from which isolates were sampled. Map drawn with the R package *rnaturalearth* (https://github.com/ropensci/ rnaturalearth) using data from Natural Earth (http://www.naturalearthdata.com/) under a CC BY license. **(B)** t-SNE plot constructed from the binary presence/absence matrix of DBLα types. Colors represent countries, each isolate is represented by a single point. Isolates with less than 20 DBLα types have been excluded. **(C)** The matching proportions obtained from the JHMM approach. An isolate's proportions are represented as a column in the graph where a column sums to one. The African isolates preferentially match with other African populations. Similarly, South American isolates match nearly entirely with other South American populations. PNG, Thailand and Iran are more closely related to the African isolates with the PNG isolates reporting a larger proportion of matching to Iran and Thailand than isolates from other countries. A small number of isolates with matching profiles that are distinct from other isolates within the same population may represent more recent migrations. **(D)** A similarity matrix indicating the mean and standard deviation of the proportions shown in Fig 2C. Origin indicates the originating country of the inferred matching proportion. The diagonal entries indicate the proportion of self-matching in each population which suggests the extent to which each population has diverged from the global set.

within Africa, but we were unable to identify within-country origins (Fig 2B). This clustering is unlikely to be the result of recent transmission as these isolates were taken from multiple geographically distant sites within countries over a period of years (Table 1). The different cluster shapes for each country are likely a result of the differences in the level of conservation of the respective DBLα types. Such shapes are also sensitive to the parameter choices of the t-SNE algorithm and should not be overinterpreted.

We found the JHMM to be the most robust approach for resolving population structure on the within and between continent scale compared to previous binary-based methods used for

analyzing *var* gene population structure (see S1 Text) discussed below. As 18,578 of the 32,682 (56.8%) DBLα types were only seen once (i.e., singletons), any approach based on binary presence/absence will ignore the majority of the sequence data available, reducing the sensitivity of the method. In contrast, the binary approach of the JHMM accounts for all the available sequences. Fig 2C displays the matching proportions from the JHMM when we compared each isolate with every other isolate, grouped by country of origin. Higher proportions of matching to one's own population (i.e., to other *P. falciparum* isolates from the same country) suggest a higher level of divergence from the remaining populations. This can be interpreted as genetic divergence of local populations within each country from the rest of the world. Gabon was identified as the most divergent African population (mean = 0.61 ± 0.15) and Colombia the most divergent South American population (mean = 0.83 ± 0.04) (Fig 2C and 2D). In addition, the French Guianan isolates appear on a curved manifold suggesting that the pairwise relationship between these isolates may be less uniform than seen in the other countries. Moreover, the relationship between countries was found to be robust to the number of isolates for each country (see Materials and Methods and S2 Fig). The geographic structure patterns by continent that we observe are in line with previous analyses using microsatellites and SNPs [30–34,46].

To investigate whether multiplicity of infection has confounded our results, we performed a multinomial logistic regression using an isolate's country of origin as the dependent variable, with the inferred proportions as well as the number of DBLα types per isolate and the disease status of the *P. falciparum*-infected individual (see Table 1) as predicting variables. While we do not infer the multiplicity of infection for each isolate directly, the number of DBLα types was found to be a weaker predictor of an isolate's country of origin than the inferred mixture proportions. After accounting for the inferred mixture proportions, the number of DBLα types was not found to be significantly associated with an isolate's country of origin (all $p > 0.97$). This is consistent with the large overlap in the distributions of the number of DBLα types by country shown in S1 Fig. If the observed signal was driven entirely by the overall number of types we would expect to see similar levels of overlap in both the t-SNE plot (Fig 2B) and the JHMM mixtures (Fig 2C), which is not the case. As isolates from Gabon, PNG, and Ghana were all from asymptomatic infections while the remaining isolates were from uncomplicated malaria cases with exception of Peru where both asymptomatic and uncomplicated malaria cases were analyzed, disease status was confounded with country and thus a small impact cannot be excluded. Despite this, clustering by country within each disease category is still evident although the distance between the major groups is reduced and the clusters are less clearly defined (S3 and S4 Figs). Some of this reduction in definition is likely to be the result of the reduced number of isolates leading to a weaker signal for the t-SNE algorithm. The t-SNE algorithm is also not guaranteed to preserve large distances and thus changes in distances between clusters cannot be easily interpreted. In addition, the JHMM inferred proportions were found to have a significant and larger effect size than disease status in the multinomial logistic regression. This coupled with the observation that the inferred proportions for Ghanaian isolates more closely resembled Ugandan isolates than the other asymptomatic isolates, suggests that the overriding signal in this dataset is due to geography.

In order to further resolve the relationships between countries, we also excluded the within-country self-matching proportions to allow for visualization of these relationships with higher resolution (S5 Fig). The African *P. falciparum* populations exhibited higher matching proportions with other African populations, and a similar pattern was observed among the South American populations (Fig 2D). The Iranian, Thai and PNG isolates were more closely related to the African isolates than to the South American isolates, which is consistent with the expansion of *P. falciparum* out of Africa toward Asia [31,33].

## South America: Out of Africa hypothesis

Our comparisons to a global DBLα type dataset and JHMM analysis allowed us to further investigate the "Out of Africa" hypothesis and build upon previous work [31,39]. We compared each South American isolate to all non-South American isolates from Africa and Asia/Oceania to determine the proportion of DBLα type matching proportions among isolates. These comparisons revealed a higher matching proportion between the South American and African isolates than with the Asian or PNG isolates (S6 Fig). These results are in line with the hypothesis that *P. falciparum* was introduced into South America from Africa through the trans-Atlantic slave trade [31].

## Evolutionary relationships to *P. praefalciparum*

To investigate the emergence and adaptation of *P. falciparum* in human populations, we used the JHMM to compare a *P. praefalciparum* isolate [5] against all other *P. falciparum* isolates in our DBLα type dataset.

The JHMM estimates the likelihood of a match between every base in each *P. praefalciparum* DBLα domain and every other DBLα domain sequence including the other *P. praefalciparum* domains. After controlling for the number of isolates and DBLα types sampled in each country, the estimated base levels probabilities were aggregated to give the estimated matching proportions at the country level. Unlike in the previous comparisons, where we normalized these proportions at the isolate level, here we normalized at the level of each DBLα type in the *P. praefalciparum* genome. This is equivalent to ignoring the overall prevalence of each DBLα type, which compensates for our analysis of only a single *P. praefalciparum* repertoire.

Fig 3 indicates that 81.5% of the estimated ancestry proportions for the *P. praefalciparum* DBLα types is found in other types within the *P. praefalciparum* repertoire. This is expected as it is a different species and thus the DBLα domains have diverged significantly from those seen in *P. falciparum*. However, 18.5% of the ancestry of *P. praefalciparum* DBLα domains matched those of *P. falciparum* DBLα types from field isolates. This is a similar proportion to that found in an analysis of *P. reichenowi* versus *P. falciparum* using laboratory clones [43]. We identified similar relationships between the *P. praefalciparum* types and the African, Asian, and PNG isolates (Fig 3). However, South American isolates appear more divergent representing only 3.67% of the inferred ancestry. The smaller proportions in South America could be attributable to stronger bottlenecks during introduction.

## Insights into the recombination structure of the *var* DBLα domain

We sought evidence for recombination "hot spots" in the sequences encoding the DBLα domain given our unique dataset of over 30,000 DBLα types collected from over 1,200 field isolates worldwide. As was also found previously [43], there did not appear to be any regions of very high or low recombination along the region of the *var* gene encoding the DBLα domain. S7 Fig illustrates the homogeneity of the recombination rate by plotting the number of recombinations at all locations inferred using the JHMM approach on the multiple sequence alignment built using Gismo [49], versus the occupancy of the alignment at that column. A sliding window of 15 columns was applied to the ratio of recombination count to alignment column occupancy but only one region (alignment columns 355–358) was found to be a mild outlier with a ratio between 1.5 and 3 times the interquartile range. Thus, this analysis supports the homogeneous recombination structure of the region of *var* genes encoding the DBLα domain identified in Zilversmit et al. (2013) [43], as well as the more recent analysis of mitotic recombinants by Claessens et al. (2014) [27] and Zhang et al. (2019) [29].

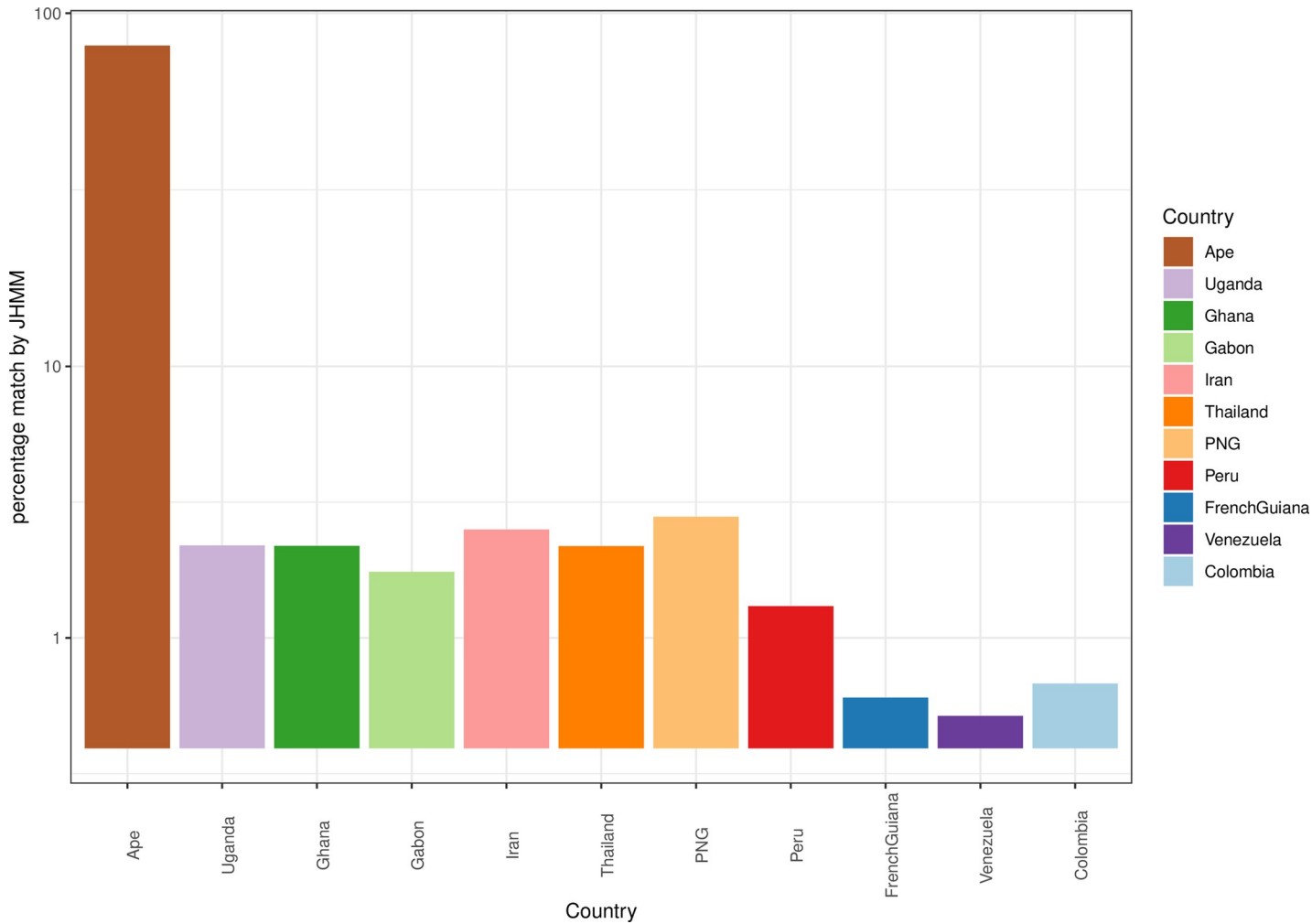

**Fig 3. The matching proportions of the *P. praefalciparum* isolate against the global *P. falciparum* populations.** The African, Asian, and PNG isolates provide the highest proportions with Gabon being the most differentiated. The South American isolates are the most distant. (Note: y-axis is in log scale).

## Comparison with previous methods

The JHMM was able to identify finer scale population structure than previous approaches. We compared our result using the JHMM approach to previous methods used for analyzing *var* gene population structure (S1 Text). Most previous methods have relied on analyzing binary presence/absence matrices of DBLα types after typically clustering at a 96% DNA sequence identity threshold. To construct such a matrix we used the pipeline described in Ruybal-Pesán-tez et al. (2017) [40]. Using this matrix, comparisons were made to the phylogeny-based approach of Tessema et al. (2015) [38] (S8 Fig) as well as the admixture approach of Rougeron et al. (2017) [39] (S9 and S10 Figs). Neither of these approaches was able to reconstruct the same level of detailed population structure as the JHMM method. As an alternative, by applying t-SNE, we were able to better resolve the global structure from the binary presence/absence matrix (Fig 2B). The t-SNE is able to distinguish structure at multiple scales in high dimensional settings while preserving local structure. In this analysis there was clear separation by country. We also considered using a BLAST based distance matrix in place of the binary presence/absence matrix, prior to visualization with t-SNE but found it provided a poorer

resolution of the global population structure (S11 Fig) [50,51]. Finally, an alternative alignment free *k-mer* based approach using Feature Frequency Profile analysis was also considered but failed to accurately distinguish all three African countries (S12–S14 Figs) [52].

## Highly conserved DBLα types at the global scale

We and others have previously shown that a number of DBLα types are conserved spatially, at the scale of countries and regions [7,37–40]. We were therefore interested in exploring whether certain DBLα types were conserved at the global scale. By examining the frequency of all 32,682 DBLα types identified in our dataset, we found that 56.8% of the DBLα types were rare (i.e., seen in only one isolate) after clustering at 96% pairwise sequence identity (Fig 4A). This agrees with previous findings from a number of studies, and is consistent with the impact of immune selection on a diverse population [7,37–42]. This pattern was largely driven by the DBLα types found among the African isolates, consistent with the overall higher diversity of *P. falciparum* populations in Africa compared to Asia/Oceania or South America [7,37–40]. S15 Fig shows the conservation of DBLα types across countries indicating that, although the majority of DBLα types were confined to a single location, many remain conserved across diverse regions with 60 types seen in all three continental areas consisting of Africa, Asia/Oceania and South America.

To further investigate these conserved types, we focused our analysis on the 100 most frequent DBLα types (i.e., those seen in > 50 isolates) to investigate the extent to which these high-frequency types were conserved over large geographic scales. We constructed a tile plot based on the presence/absence of each high-frequency type across all isolates and clustered the DBLα types using the squared correlation between their presence/absence to generate a similarity matrix (Fig 4B). Thus, conserved types that either often appear together or nearly always appear separately will be clustered closer together. In addition, each DBLα type was annotated with its most likely major DBLα domain (DBLα 0, 1, 2) as well as being further classified based on their upstream promoter sequences (ups) as either upsA or upsB/upsC (i.e., non-upsA) using the method described in Ruybal-Pesántez et al. (2017) [40].

Twenty-two percent of the high-frequency types were found in isolates from African, Asian and South American populations. The patterns of geographic population structure were also observed for the high-frequency DBLα types with a cluster of predominantly South American types evident in Fig 4B. The presence of a distinct South American cluster is consistent with the identification of a limited pool of DBLα types in the Brazilian Amazon by Albrecht et al. (2010) [53]. When we annotated the high-frequency types based on their DBLα domain, 25% matched most closely to DBLα1 suggesting they were upsA type *var* genes (Fig 4B). The proportion of DBLα1 types conserved across countries reflects their expected proportion per genome [6], indicating they are not over- or under-represented on a geographical scale. We found evidence for geographic variation in these types and we reason that local adaptation of *P. falciparum* populations to distinct selective pressures, vector populations, and hosts may have played a role in shaping these patterns [39,41].

## Comparative analyses of highly conserved DBLα types to previously published data

In an attempt to annotate these high-frequency types, we also searched each sequence against the NCBI nucleotide reference database [50,54,55], as well as the known conserved *var* types that encode DBLα domains: *var*1 and *var*3 [6,56,57]. Any matches greater than 96% pairwise sequence identity are reported in S1 Data.

The high conservation of our 100 high-frequency DBLα types was also confirmed by searching for them in a recent independent global assembly of *var* genes conducted by Otto

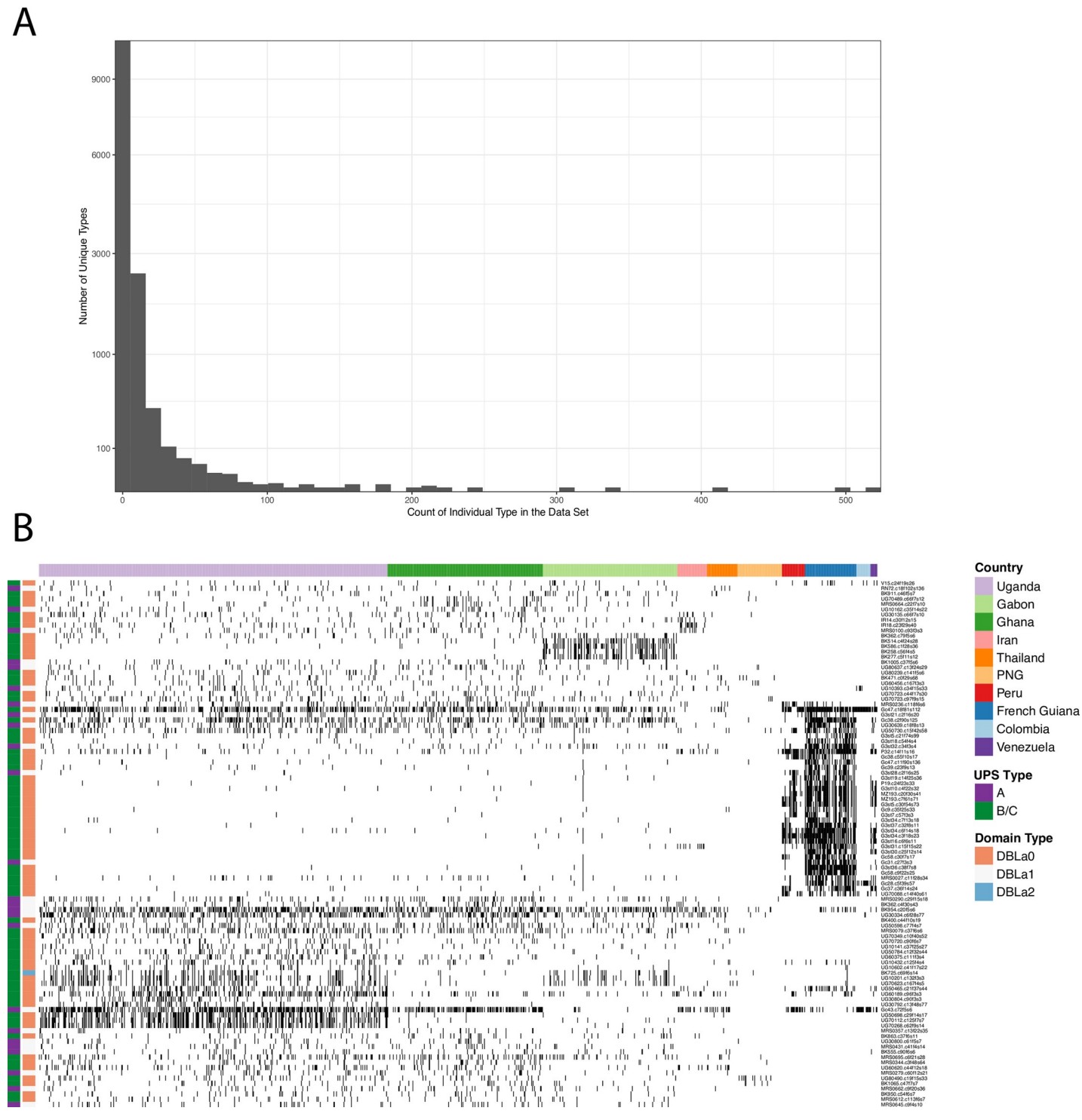

**Fig 4.** **(A)** A histogram of the frequency with which DBLα types appear in the dataset after clustering at the 96% identity level. **(B)** A tile plot representing the binary presence/absence matrix of the top 100 most conserved DBLα types in the dataset. Countries are distinguished by the top colored row while the first two columns indicate each DBLα type annotated (colored) with its most likely ups type (A, B/C) and major DBLα domain (DBLα 0, 1, 2), respectively. The South American countries are evident on the right-hand side of the plot as are a number of Gabonese types that appear almost exclusively in Gabon.

et al. (2019) [5]. This global analysis included countries from around the world except for South America. We found matches to six countries in Asia (Bangladesh, Cambodia, Laos, Myanmar, Thailand and Vietnam) and eight countries in Africa (Democratic Republic of Congo, Gambia, Ghana, Guinea, Kenya, Malawi, Mali and Nigeria) (S16 Fig). The types observed at the highest frequency in our dataset were also observed in a high number of countries across the world, with the median number of countries with matches being 11 and ranging from 1 to 14 (S16 Fig). Of the 100 high-frequency DBLα types, 81 were seen in both Asia and Africa, with 19 types that were found only in African isolates. For the complete list of countries with matches to each high-frequency type, along with the corresponding BLAST results, see S2 and S3 Data, respectively.

The most conserved DBLα type (seen in 521 isolates, 46.2% of our global dataset) was seen in every one of our study countries except PNG. Additionally, BLAST hits to this type were found in other published *var* gene sequences from Brazil, Kenya, Tanzania and Malawi, further demonstrating its conservation in *P. falciparum* populations from other African and South American countries not included in this study. In the Otto et al. (2019) [5] study, this "globally-conserved" DBLα type was homologous to *var* genes carrying the semi-conserved structure NTSB3-DBLα0.9/0.11/0.16-CIDRα2.1/2.4/3.4 (S1 Table). The occurrence of this sequence across multiple group B/C configurations indicates that it is a conserved sequence block shared by multiple DBLα sequences and *var* genes and is thus collapsing multiple DBLα sequences. Otto et al. (2019) have identified such conserved *var* genes [5]. The second, third, fifth, 11th and 22nd most conserved DBLα types were annotated as the well-known, conserved *var*1 genes [6,7,56] that are often truncated and that have unknown function [6]. None of the 100 high-frequency DBLα types matched within 96% pairwise sequence identity to the conserved *var*3 genes of Rask et al. (2010) [6]. This was expected as the universal DBLα primers used (developed by Taylor et al. (2000) [58]) do not match the distinct DBLα1.3 domain of *var3* which is a DBLα-DBLζ hybrid [6].

The 100 high-frequency types also included homologues of the conserved DBLα sequences associated with selective sweeps of alleles associated with antimalarial resistance on chromosomes 4, 6 and 7 (S1 Table) [5]. Other conserved arrangements of domain subtypes identified in Otto et al. (2019) [5] were also associated with the 100 high-frequency DBLα types (S1 Table). These included features previously associated with functional phenotypes, e.g. CIDRα1, which binds to endothelial protein C receptor in severe malaria [21], DBLβ which binds ICAM1 in severe malaria [17,59,60], domain cassettes 8 and 5, which were expressed in severe malaria in Africa [18], and domain cassette 9, which was expressed in severe malaria in Papua [14]. However, other highly conserved structures have not been previously investigated, for example the fourth most frequent DBLα type was associated with the highly-conserved PfEMP1 structure NTSB3-DBLα0.4-CIDRα6-DBLβ5-DBLγ10-DBLδ6-CIDRβ2. The high frequency and conservation of these 100 DBLα types in our study and their association with conserved *var* gene structures suggests these genes may have an important biological function and thus warrant further examination.

## Discussion

By analyzing the geographic relatedness of a large dataset of *P. falciparum* field isolates from 23 locations in 10 countries, we demonstrate that the evolution of the parasite population emerging "Out of Africa" underlies current patterns of DBLα type diversity. This evolutionary result presents an opportunity for contemporary malaria surveillance in these times of globalization. Human migration due to conflict, food security, economic opportunity, will undoubtedly perturb the observed geographic patterns.

Specifically, in identifying geographic population structure in DBLα types of the major variant surface antigen of *P. falciparum* blood stages, we provide compelling evidence for use of these types in malaria surveillance. Based on the JHMM output, we can track DBLα types to specific localities, countries and continents to monitor changing patterns of malaria transmission. Of significance, we can see geographic population structure of these types within Africa, the origin of *P. falciparum* and where high levels of diversity still exist in contemporary *P. falciparum* populations. Our data obtained by targeted amplicon sequencing of a PCR of a 450bp fragment of the *var* multigene family encoding the DBLα domain presents a relatively cost-effective method to reveal global population structure in the genes encoding the major surface antigen in comparison to using entire *var* genes. The DBLα domain has been shown to encode variant-specific epitopes recognized by host antibodies in an age-specific manner [20,61,62]. Such antibodies have been shown to regulate parasite density and protect against clinical disease [61]. Thus, DBLα types can also be used as potential markers of geographic patterns of variant-specific population immunity in relation to location-related exposure to specific PfEMP1 variants.

In order to identify geographic population structure in the DBLα domain we show that utilizing methods designed specifically for the investigation of *var* gene evolution provides substantially more insight than more naive approaches such as a binary matrix of presence/absence of types. Using the JHMM method [43], we were able to distinguish all countries within the global dataset. We describe multiple sub-populations in South America consistent with a previous analysis [39], and further support the "Out of Africa" hypothesis where *P. falciparum* was introduced into South America from Africa, likely due to the trans-Atlantic slave trade [31]. Interestingly, we found that the South American isolates were not more or less related to any isolates from any of the African countries, which may indicate that we have not sampled DBLα types from the "origin" populations out of Africa, or that the South American sequences have diverged significantly from African sequences due to e.g. different ecological and host niches.

We provide compelling evidence for DBLα type population structure within Africa with Uganda, Ghana and Gabon showing distinct matching proportion profiles using the JHMM method. Gabon was found to have diverged further from the other two African countries and this was supported by a comparison with a *P. praefalciparum* isolate. The JHMM proportions suggest the *var* populations in Asia/Oceania more closely resemble African populations than those seen in South America. This is consistent with the expansion of *P. falciparum* out of Africa toward Asia [31,33]. Overall, our results strongly support geographic variation of DBLα types on a continental scale.

A number of highly conserved DBLα types were also observed to occur globally. Some of these matched the previously identified *var*1 type but many have not been well described. Whilst we cannot ascertain their function, the high prevalence and strong conservation of these DBLα types on a global scale could indicate they have an important biological function and warrant further attention. Biological function is not the only explanation for conservation of some of these DBLα types. For example, eight of these 100 conserved DBLα types were associated with selective sweeps on chromosomes 4 and 7 of alleles that confer antimalarial resistance to pyrimethamine and chloroquine, respectively [5]. The conserved *var* gene associated with a selective sweep of four *var* genes in the subtelomeric region of chromosome 6 by Otto et al. (2019) [5] and related DBLα type conserved in our dataset has no apparent association with antimalarial resistance. Other as yet unidentified selective sweeps of alleles advantageous to the parasite could be responsible for some of the conserved DBLα types.

We have demonstrated the power of using the JHMM method to investigate *var* gene populations. However, the method is computationally expensive and can take days to run on large

datasets. With the increasing size of datasets and for this approach to be useful in larger geographic surveillance studies, its computational performance will need to be improved. A large database of geographically assigned DBLα types analyzed by the JHMM for geographic signatures would be the goal to underpin a cost-effective surveillance method in a manner similar to influenza global surveillance with hemagglutinin sequences.

In describing the global population structure of *P. falciparum* DBLα types and their relationship to other *Plasmodium* species, we have demonstrated the geographic variability of *var* population structure despite the incredible diversity and high recombination rate. This has important consequences for the use of DBLα types in the surveillance of *P. falciparum* as well as for the assessment of population immunity to specific PfEMP1 variants. This study identifies current patterns of DBLα type diversity as a baseline from where changes can be assessed by sequencing the amplicons of a single PCR. Moreover, changes in patterns of global mobility that may involve mixing of populations not previously co-located would have significant consequences for contemporary *var* evolution and immune evasion. Such gene flow could lead to epidemics of genomes containing geographically novel *var* genes in endemic populations lacking specific immunity. Hence routine malaria molecular surveillance can be expanded from the current use of neutral markers such as SNPs and microsatellites to include a marker under immune selection that also identifies geographic origin.

## Materials and methods

### Ethics statement

The study was reviewed and approved by the ethics committee at the University of Melbourne, Australia (approvals #HREC 144–1714 and #HREC 195–5645).

### Jumping hidden Markov model

We used the implementation of the jumping hidden Markov alignment model (JHMM) (Mosaic) algorithm kindly provided by Zilversmit et al. (2013) [43] to estimate the posterior likelihood of each unique DBLα type being related at each position to any other type in the dataset after first translating the centroid sequence for each DBLα type into its corresponding protein sequence. Here, the centroid is defined using the USEARCH clustering algorithm with all sequences in a DBLα type being within 96% pairwise identity of the centroid.

Briefly, the model combines the pair-HMM model of pairwise sequence alignment [63] with the probabilistic model of Li and Stephens (2003) [64], which describes the impact of recombination on haplotype sequence diversity. The resulting model is very similar to that used in *fineStructure* [65]. Given a set of n source sequences, a target sequence is aligned by choosing a starting point uniformly from all sites in the source sequences. The alignment starts in a match state with probability $\pi_m$, and in an insert state with probability *I*. The alignment is constructed by exploring the space of match, insert and delete spaces similar to a standard pair-HMM. At each step there is a probability of jumping between source sequences (either to a match or insert state) to allow for recombination. The most likely path through the search space is found using the Viterbi algorithm, whilst the posterior probabilities of alignment at each location are found using the forward-backward algorithm. A more thorough description of the algorithm is given in the supplementary material of Zilversmit et al. (2013) [43]. The exact commands used to run the Mosaic algorithm are provided in the "supplementary_-methods_2.Rmd" file on GitHub, along with the the Mosaic source code, at: https://github.com/gtonkinhill/global_*var*_manuscript.

Due to the large number of DBLα types in our dataset, a number of steps had to be taken to deal with the increased computational complexity involved. Initially the DNA sequences from

the centroid of each DBLα type were translated into their respective protein sequence and removed if the resulting protein sequence contained a stop codon. Alignment of protein sequences has previously been found to be more sensitive for the analysis of *var* genes [14]. The DNA sequences that could be translated were then clustered at 96% identity using USEARCH as described previously [66]. The protein sequences that corresponded to the centroids of these clusters were used in the JHMM after trimming to a consensus alignment using Gismo [49].

To train the non-jump parameters of the model, we implemented a script to run the Viterbi training algorithm [63] with the jump probability set to "0". This was used in place of the Baum-Welch algorithm used by Zilversmit et al. (2013) [43] as it is more efficient and it was not feasible to run the Baum-Welch algorithm on our dataset. The non-jump parameters were then fixed, and a composite-likelihood surface was generated for the jump parameter by searching a randomly selected subset of 1000 sequences against the entire dataset. This significantly reduced the computational time required. The maximum-composite-likelihood estimate for the jump parameter was then used in the analysis.

As each translated DBLα type was only represented once in the model, we had to account for this when constructing the expected country mixture proportions for each isolate. The proportion of each isolate that came from a particular country was found by summing the posterior likelihood that an amino acid position originated from a sequence in that country and normalizing by the total sequence length of that isolate. In more detail, let $R_t$ be the set of DBLα type centroids in the target isolate (T), $R_S$ the set of DBLα type centroids in an alternative isolate (S), $r_{tl}$ be the l$^{\text{th}}$ amino acid in a centroid t from the target isolate, $L_{r_t}$ the length of centroid $r_t$, k be the set of all isolates in country k and $X_{TS}$ be the proportion of amino acids inherited by T from S. Then,

$$E[X_{TS}] = \frac{\sum_{r_t \in R_T} \sum_{l \in L_{r_t}} \sum_{r_s \in R_S} P(r_{tl} \text{ is from } r_s)}{\sum_{r_t \in R_T} L_{r_t}}$$

Here, if the target centroid was a singleton, $P(r_{tl} \text{ is from } r_s)$ is taken as the posterior probability identified from the JHMM algorithm normalized by the number of times the centroid S was found in the full dataset. If the target DBLα type was found multiple times in the dataset, $P(r_{tl} \text{ is from } r_s) = 0$ if the DBLα types are not identical and 1/(*number of identical centroids*) otherwise. The algorithm was tested with identical DBLα types present and found to always split the posterior probability evenly between the identical copies. Finally, the expected proportion of an isolate most closely related to a country was found by averaging over the respective matches to isolates in that country. That is,

$$Average\ expected\ proportion\ from\ country\ k = \frac{\Sigma_{S \in k} E[X_{TS}]}{k}$$

To test the robustness of the resulting proportions to the number of isolates sampled from each country, we repeatedly randomly subsampled 10 isolates from each country and recalculated the proportions. Ten isolates were chosen as this was the minimum number of isolates found in a single country (Venezuela). S2 Fig indicates that the resulting proportions indicate similar relationships between countries. The standard deviations of these proportions were higher than in the full analysis which is as expected given that the full analysis involved many more isolates.

### Binary analysis

The DBLα sequences were clustered using a pipeline based on the USEARCH v8.1.1831 software suite [66] as described previously in Ruybal-Pesántez et al. (2017) [40]. Specifically, the

*derep* prefix command was used to sort the sequences based on the number of duplicates present before redundant sequences were removed. The remaining sequences were then clustered using the cluster_fast command at 96% pairwise identity.

A binary presence/absence table was generated by searching the original sequences against the centroids from the clustering using the usearch_global command. The resulting matrix was forced to be binary by setting any entry greater than "0" to "1". Isolates were removed if they had less than 20 DBLα types and remaining singleton types were removed from the matrix. This threshold was chosen as it has been found previously to filter out poor quality isolates [39,40]. To test the robustness of our results to this threshold, the binary analysis was repeated with a threshold of two DBLα types. The resulting t-SNE plot is given in S17 Fig and indicates that the population structure is still clear. The t-SNE and principal component analysis (PCA) [48,67] were performed using R [68]. The matrix was then converted into the format expected by Admixture v1.3 [69] where a present type was represented as an alternative allele in a haploid chromosome. Finally, a phylogenetic tree was constructed by treating each isolate's corresponding row in the matrix as its binary sequence. RAxML v8.2.8 was then run using the BINCAT model. The code as well as a brief description of the methods is available on GitHub at: https://github.com/gtonkinhill/global_*var*_manuscript.

## Feature Frequency Profile (FFP)

In the Feature Frequency Profile (FFP) analysis, we used a reduced RY alphabet to describe the gene sequences for each isolate, where R stands for the purine bases (AG) and Y stands for the pyrimidine bases (TC). This alphabet was chosen to reduce memory usage as suggested in Sims et al. (2009) [52]. To choose an appropriate *k-mer* length, we first looked at word usage to obtain a lower bound. As suggested in the FFP manual, we counted the number of times *k-mers* appeared at least twice in the dataset for different values of k. The peak of this distribution occurred at a length of 17 (S13 Fig). *K-mers* shorter than this are very commonly found and thus offer little additional distinguishing information. By investigating the relative entropy between the observed frequency of a *k-mer* and a *k-2* Markov model [52], we can also obtain an upper bound for the *k-mer* length. When the relative entropy is small, this indicates we can predict the frequency of *k-mers* from smaller *k-mers* [52]. Consequently, this gives an upper bound on k which we found to be approximately 22 (S14 Fig). Thus, a choice of k = 20 was sensible.

The frequency of all 20-mers was then calculated for each isolate after converting their sequences into the reduced RY alphabet. A distance matrix was generated using Jensen-Shannon divergence. The FFP v3.19 program was used to estimate the upper and lower bounds for k, based on the 3D7 isolate sequence. The distance matrix was then generated using a custom Python script, before FastME v2.1.4 [70] with default settings was run to produce a neighbor-joining tree. The final tree diagram was generated using the R package *ggtree* [71].

## Comparison to a previously published global assembly of *var* genes

The 100 high-frequency DBLα types identified in this study were used to BLAST query a previously published global assembly of *var* genes [5]. The predicted PfEMP1 domain structure of the hit and the subject sequences were extracted and ranked by E-value (*Note*. All low E-value sequences were visually inspected for conservation). Conserved structures are indicated in S1 Table. The accession numbers of the conserved assembled *var* genes that occurred in clusters associated with selective sweeps for regions of chromosomes 4, 6, 7, 8, and 12 were obtained from the author [5]. DBLα types that had a BLAST subject hit that was contained in these selective sweep clusters were identified and are indicated in S1 Table.

## Data

All DBLα sequence data included in this global study were obtained from *P. falciparum* isolates previously collected and published as described in Table 1 [31,36,39–41] except for the *P. falciparum* isolates from Peru (Zungarococha/Mazan) [46], Thailand [31,47], and Iran [31] that were previously collected but were sequenced for the present study. For these *P. falciparum* isolates, PCR amplification of the DBLα domain and sequencing on a 454 platform (Roche) was performed following the same protocol as we have previously published [36,39–41,72]. The *P. falciparum* isolates from Papua New Guinea (PNG) were processed using a different protocol and Sanger sequencing as described in Tessema et al. (2015) [38]. Apart from the *P. falciparum* isolates from PNG, the 454 DBLα sequence data was processed using the same bioinformatic pipeline described in Rask et al. (2016) [72] and Ruybal-Pesántez et al. (2017) [40]. All data to reproduce this analysis is available along with the code on GitHub at: https://github.com/gtonkinhill/global_*var*_manuscript. Further details on the *P. falciparum* isolates included from each region are described below.

## South America

The 128 uncomplicated *P. falciparum* isolates were collected between 2002 and 2008 from various locations across South America (Colombia, Venezuela, French Guiana, Peru) and are further described in Restrepo et al. (2008), Yalcindag et al. (2012), and Rougeron et al. (2017) [31,39,73] as well as 13 asymptomatic *P. falciparum* isolates from Peru (Zungarococha/Mazan) as described in Branch et al. (2011) [46].

## Africa

**Gabon.** The 201 asymptomatic *P. falciparum* isolates were collected in 2000 from Bakoumba, Gabon and are further described in Ntoumi et al. (2002), Fowkes et al. (2006), and Day et al. (2017) [41,74,75].

**Uganda.** The 517 uncomplicated *P. falciparum* isolates were collected from six sentinel sites across Uganda between 2006–2007 and are further described in Hopkins et al. (2008) and Ruybal-Pesántez et al. (2017) [40,76].

**Ghana.** The 231 asymptomatic *P. falciparum* isolates were collected from two catchment areas in Ghana in 2012 and are further described in Ruybal-Pesántez et al. (2017) and Rorick et al. (2018) [36,77].

## Asia/Oceania

**Thailand.** 46 uncomplicated *P. falciparum* isolates were collected between 2005–2007 from various sites in Thailand and are further described in Pumpaibool et al. (2009) and Yalcindag et al. (2012) [31,39].

**Iran.** 45 uncomplicated *P. falciparum* isolates were collected between 2000–2003 from Iran and are further described in Yalcindag et al. (2012) [31].

**PNG.** In contrast to the previous datasets, the sequences obtained from PNG were not processed using the same bioinformatic pipeline. The PNG sequences were generated using Sanger sequencing after first amplifying for DBLα domains using the same DBLα primers as were adapted for the 454 sequencing [37]. The isolates were sampled from two geographically distinct areas in PNG (Wosera/Mugil) and resulted in *var* DBLα sequences from 33 and 35 isolates, respectively. A more detailed description of the experiment, sampling and bioinformatic pipeline can be found in Tessema et al. (2015) [38].

## Supporting information

**S1 Text. Additional details comparing the JHMM approach to previous methods used for analyzing *var* DBLα population structure (S18 and S19 Figs).**
(PDF)

**S1 Fig. Box plots representing the number of unique DBLα types per isolate in each country.** The African countries have significantly higher numbers of types indicating the higher prevalence of multiple-genome infections in Africa. Isolates with less than 20 DBLα types have been excluded.
(TIF)

**S2 Fig. A heatmap indicating the mean and standard deviation of matching proportions inferred using the JHMM after repeatedly sub-sampling 10 isolates from each country.** The relationship between countries mirrors that seen in Fig 4B indicating that the result is robust to the sampling coverage for each country.
(TIF)

**S3 Fig. t-SNE plot constructed from the binary presence/absence matrix of DBLα types in the isolates from Uganda, South America, Thailand, and Iran from uncomplicated malaria cases.**
(TIF)

**S4 Fig. t-SNE plot constructed from the binary presence/absence matrix of DBLα types in the isolates from Gabon, Ghana, Peru, and PNG from asymptomatic malaria cases.**
(TIF)

**S5 Fig. The matching proportions obtained from the JHMM approach where the self-matching proportions have been removed to make the between/among country comparisons clearer.** An isolate's proportions are represented as a column in the graph where a column would add to one if self-matching was included. The African isolates preferentially match with other African populations. Similarly, South American isolates match nearly entirely with other South American populations. PNG, Thailand and Iran are more closely related to the African isolates with the PNG isolates reporting a larger proportion of matching to Iran and Thailand than isolates from other countries. A small number of isolates with matching profiles that are distinct from other isolates within the same population may represent more recent migrations.
(TIF)

**S6 Fig. The matching proportions obtained after searching South American isolates against the global database excluding the South American isolates.** This prevents the algorithm from assigning ancestry to other South American isolates and thus allows us to focus on the relationships with the remaining countries. The proportions indicate no strong link between any of the South American countries and any one African country.
(TIF)

**S7 Fig. The top bar plot indicates the occupancy of each column of the Gismo multiple sequence alignment while the bottom bar plot indicates the number of jumps that were inferred to occur at that location from JHMM model.** The symmetry between the two plots indicates that recombination occurs throughout the DBLα tag with only one multiple sequence alignment column found to be an outlier. An alignment of the relevant homology blocks from Rask et al. (2010) [6], is given below the two bar plots.
(TIF)

**S8 Fig. A phylogenetic tree built using RaxML [78] with the BINCAT model and treating each isolate's binary presence/absence vector as a binary sequence.** The population structure evident in the t-SNE plot is reproduced in this analysis. Peru is split into two populations and Ghana is more distinct from Uganda than the FFP and Admixture analysis. Colombia is found to be closer to the other South American isolates in this analysis.
(TIF)

**S9 Fig. A bar plot indicating the matching proportions inferred from Admixture [69] with two latent populations.** An isolate is represented as a single haploid chromosome with the alternative allele indicating that a DBLα type is present in that isolate. The separation between African and the non-African populations is clear.
(TIF)

**S10 Fig. The cross-validation error for different values of K (the number of latent clusters) when running Admixture on the binary type matrix.**
(TIF)

**S11 Fig. A t-SNE plot generated using a BLAST based pairwise distance matrix [51].** Whilst clustering by country is evident, the resolution is poorer than was achieved using the binary presence/absence-based distance.
(TIF)

**S12 Fig. An unrooted neighbor-joining tree.** The tree was constructed using the default Fas-tMe v2.1.4 [70] method from a distance matrix generated using the Feature Frequency Profile (FFP) approach of Sims et al. (2009) [52] with a *k-mer* length of 20. The country level population structure is evident; however, Ghana is less separated from Uganda than in the t-SNE and JHMM approaches.
(TIF)

**S13 Fig. A plot of the *k-mer* vocabulary size (the number of *k-mers* seen at least twice) versus *k-mer* length.** This can be used to set a lower bound for the choice of *k-mer* length by looking for the maximum of the vocabulary size [52].
(TIF)

**S14 Fig. A plot of cumulative relative entropy (CRE) versus *k-mer* length which can be used to set an upper bound on the *k-mer* length by selecting the point when the CRE approaches zero.** See Sims et al. (2009) [52] for a detailed description.
(TIF)

**S15 Fig. A scatter plot of the number of times each DBLα type was identified in an isolate versus the number of countries it was found in.** The high density of the types seen in only one country is driven by the large number of unique DBLα types identified.
(TIF)

**S16 Fig.  A.** The frequency of the 100 high-frequency DBLα types in our global dataset (i.e., the number of *P. falciparum* isolates each type was observed in out of 1,248 isolates). **B.** The presence/absence of each high-frequency type after searching for them in the independent assembly of *var* genes from Otto et al. (2019) [5], where black denotes presence and white denotes absence stratified by country of origin. The order of DBLα types along the x-axis is the same in both A and B.
(TIF)

**S17 Fig. A t-SNE plot after only filtering out isolates with less than two DBLα types.** Whilst the clustering is less defined than Fig 2B, the overall grouping by country is still clearly evident suggesting that the result is robust to the commonly used practice of filtering out isolates with less than 20 DBLα types.
(TIF)

**S18 Fig. The first two components after performing a Principal Component Analysis (PCA) on the binary presence/absence matrix of DBLα types.** A clear separation between the South American isolates is apparent.
(TIF)

**S19 Fig. The third and fourth components from the PCA on the binary presence/absence matrix of DBLα types.** Although there is still significant overlap, the separation between the African countries is shown. A much clearer distinction was found in the t-SNE analysis.
(TIF)

**S1 Table. The accession numbers for the 100 high-frequency DBLα types and their number of occurrences in the dataset (i.e., counts).** These high-frequency DBLα types were each used to BLAST query a globally assembled *var* gene database [5] and representative hit subject sequences with the BLAST results are included along with the domain structure arrangement of the representative hit and other high identity hits, e.g. the structure NTSB3-DBLα0.2-CIDRα3.3- DBLγ1- CIDRβ1 indicates that most of the highly ranked hits had this exact structure so the gene was conserved whereas NTSB3-DBLα0.9/0.11/0.16-CIDRα2.1/2.4/3.4 indicates that three DBLα types and three CIDRα types were represented in the highly ranked hits and therefore this DBLα type was not part of a highly conserved gene structure. Conserved structures previously identified domain cassettes (DC) [6] and their association with disease are indicated. The global *var* assembly included clusters of highly conserved *var* genes that were associated with selective sweeps on chromosomes 4, 6 and 7. Any of the high-frequency DBLα types that were homologues of genes from these clusters are indicated (selective sweep associated chromosome) and a member of the cluster from an assembled *P. falciparum* genome that was used to assign the chromosome to the cluster is included along with positional information relative to other assembled genome homologues of types associated with the sweeps that were also identified in the current study.
(CSV)

**S1 Data. This table indicates all the BLAST results from comparing the top 100 most conserved DBLα sequences against the NCBI database.** The columns represent in order: the sequence identifier (ID), the sequence identifier from the NCBI database (NCBI_ID); the pairwise sequence identity of the match (pwid); the length of the match (length); the number of mismatches (n_mismatches); the number of gap openings (n_gap); the start of the alignment in the query (q_start); the end of the alignment in the query (q_end); the start of the alignment on the NCBI matched sequence (t_start); the end of the alignment on the NCBI matched sequence (t_end); the BLAST e-value (evalue); and the BLAST bit score (score).
(CSV)

**S2 Data. The table indicates the total number of countries for each of the top 100 most conserved DBLα sequences, including both the current dataset and the BLAST analysis against the NCBI database.**
(CSV)

**S3 Data. The top 100 most conserved DBLα sequences.** The columns represent in order: the sequence identifier (SeqID); the nucleotide sequence (NucSeq); a shortened version of the

sequence identifier (shortID); the number of BLAST hits to reach 96% pairwise sequence identity (Hits); the countries the sequence was found in from the current dataset (Countries_binary); the additional countries the sequence was found in through the BLAST analysis (Countries_blast); whether any of the common BLAST hits were annotated as pseudogenes (Annotation) or *var*1/2/3 (Blast against var123); and finally the coverage (coverage), pairwise identity (percentID), and e-value of the best BLAST hit (e-value).
(CSV)

## Acknowledgments

We are grateful to participants of malaria disease surveillance studies whose samples have been analyzed in this study. We appreciate the support of the Walter and Eliza Hall Institute of Medical Research for computational resources. Finally, we thank everyone involved for their continued patience as this research was disrupted due to Hurricane Sandy (New York, NY; October 29, 2012).

## Author Contributions

**Conceptualization:** Karen P. Day.

**Data curation:** Gerry Tonkin-Hill, Shazia Ruybal-Pesántez.

**Formal analysis:** Gerry Tonkin-Hill, Shazia Ruybal-Pesántez, Yao-ban Chan, Karen P. Day.

**Funding acquisition:** Karen P. Day.

**Investigation:** Shazia Ruybal-Pesántez, Kathryn E. Tiedje, Virginie Rougeron, Michael F. Duffy, Sedigheh Zakeri, Pongchai Harnyuttanakorn, OraLee H. Branch, Lastenia Ruiz-Mesía, Franck Prugnolle, Yao-ban Chan, Karen P. Day.

**Methodology:** Gerry Tonkin-Hill, Thomas S. Rask.

**Project administration:** Karen P. Day.

**Resources:** Shazia Ruybal-Pesántez, Kathryn E. Tiedje, Virginie Rougeron, Sedigheh Zakeri, Tepanata Pumpaibool, Pongchai Harnyuttanakorn, OraLee H. Branch, Lastenia Ruiz-Mesía, Franck Prugnolle, Karen P. Day.

**Supervision:** Anthony T. Papenfuss, Karen P. Day.

**Validation:** Gerry Tonkin-Hill.

**Visualization:** Gerry Tonkin-Hill, Shazia Ruybal-Pesántez, Michael F. Duffy.

**Writing – original draft:** Gerry Tonkin-Hill.

**Writing – review & editing:** Shazia Ruybal-Pesántez, Kathryn E. Tiedje, Michael F. Duffy, Yao-ban Chan, Karen P. Day.

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
