## [Decision Letter · Decision Letter 0]

21 Jul 2020

Dear Dr Day,

Thank you very much for submitting your Research Article entitled 'Evolutionary analyses of the major variant surface antigen-encoding genes reveal population structure of  Plasmodium falciparum  within and between continents' to PLOS Genetics. Your manuscript was fully evaluated at the editorial level and by independent peer reviewers. The reviewers found that your study brings novel insight into the var gene family, although several population genetics studies analysing var genes have already been published.  They appreciated the novel logarithm and the finding of population-specific signatures, which was thought to be not possible. However, both reviewers also had several concerns and critics with respect to some of the conclusions drawn and asked for a better explanation of the method. Furthermore, as the JHMM method is key to this paper it needs to be submitted to the git-hub page, and needs to be explained so it can be used also by others. The author should put the command how they run the program in the methods to allow reproducing the data and make it accessible to others in the field.The supporting datafiles, need to be better documented. Based on the reviews, we will not be able to accept this version of the manuscript, but we would be willing to review again a much-revised version. We cannot, of course, promise publication at that time.

If you decide to revise the manuscript for further consideration at PLOS Genetics, please aim to resubmit within the next 60 days, unless it will take extra time to address the concerns of the reviewers, in which case we would appreciate an expected resubmission date by email to plosgenetics@plos.org.

[LINK]

We are sorry that we cannot be more positive about your manuscript at this stage. Please do not hesitate to contact us if you have any concerns or questions.

Yours sincerely,

Carmen Buchrieser

Associate Editor

PLOS Genetics

Hua Tang

Section Editor: Natural Variation

PLOS Genetics

Reviewer's Responses to Questions

**Comments to the Authors:**

Reviewer #1: In this paper, the authors analyse the global diversity of the DBLa domains, from the polymorphic var gene family in P. falciparum. It is a huge body of work. Published data are complemented with novel sequences, and an HMMer approach was used to find distant similarity between the domains.

These findings allowed the author to find population-specific signatures, which was thought to be not possible due to the sheer diversity of the var genes. But there must be immune pressure to form "populations".

The authors use this to confirm existing population structure, analysis samples from related chimpanzee samples and speculate about the out of Africa hypothesis.

Overall I enjoyed reading the manuscript, and I have following points to discuss

The JHMM method is key to this paper. And it is described in the paper, but I would not be able to implement it from the article. Also, I am not sure if it is on the git-hub page, which is not very self-explanatory - but helpful. So I wonder if the author could put the command how they run the program in the methods to be able to reproduce the data and make it accessible to others in the field.

One argument of this study is to use PCR of DBLa to do population studies. How does that compare to SNP arrays or other PCR based methods, in terms of time and cost?

I have issues with the use of herd immunity as a term for var genes. Although the parasite is around for so long, no herd immunity was seen and so far all ideas to generate vaccines based on var genes failed. Could you please comment?

It has recently shown that P. gaboni does not have functional DBLa domains, but these are pseudonised domains, which also explains the low number. Should they not be excluded from the analysis?

I like the t-SNE plot but struggle with the binary presence/absence. In Fig 2B, some populations have a very different pattern. Could it be that there are sampling biases like two samples came from related infections that bias this analysis? Would it not be better to look for a blast based similarity matrix and do the T-SNE plot from that matrix? So I cannot understand that Colombia has a similar pattern than Ghana, and they are also more close.

Due to the Jumping HMM, the base information gets lots. Would it be possible to generate like GSEA like plots, where 10 DBLa domains get taken, and then the hits get ordered by similarity and coloured below by country as a bar code? This would give the reader a better understanding of how clear the clustering per country/region is based on individual DBLa. Ten randomly picked domains from Pf3D7 could do this plot?

Line 246, I cannot follow how the regression addresses the multiplicity of infections.

I was wondering if the JHMM could bring more population resolution in West Africa. I agree with the authors that so far their approach confirmed existing population structure. But SNP based methods struggle to find differences in West Africa. Would it be possible to use the DBLa from reference 34 and look for novel structure?

Could point eight help to narrow down the origin from the out of Africa hypothesis.

I like the evolutionary relationship to the Laverania, but why was not P. praefalciaparum used? It is the closest common ancestor and there is a pacbio assembly in the public domain.

Different methods were used to analyse the data. But was a blast approach used? I just looked at some of the data of the authors, and a simple blast of the 3D7 domains shows the proximity of African domains.

I like supporting datafiles, but they need a bit of cleaning and better documentation, like table legends etc.

I personally don't like personal communication of open access publications...

The methods should be revisited. Several terms are not defined like centroid - in which context, in the alignment? Suddenly which from DBLa to reads. What is OTU, FFP? The link to the different program on the git-hub would be useful.

Minor points

There are newer references to confirm 60 var genes in Pf, like the recent papers on Pacbio assemblies or the reference 34 - line 83

Could you please provide a reference for the expression of var genes in early gam? Line 90

Line 114 42% amino acid similarity

Struggle to understand line 237ff, Gabon most self-matching or divergent - is that not contradictive?

Line 309 - is it possible to distinguish between mitotic and meiotic recombination? So far the first was just seen in vitro.

line 405 CIDRa1.n n?

Some figures need higher resolutions.

There are some discussion about speed. How long does it take to do the JHMM analysis? Hours, days?

Reviewer #2: Tonkin-Hill et al sought to understand P. falciparum population genetics using the DBLa domains from the var gene family. These ~400bp domains are extremely polymorphic, making them a good candidate as a ‘barcode’ to define each P. falciparum isolate. The concept has been utilised multiple times by this lab, however here they present the first worldwide analysis (albeit limited to 10 countries), using 125 ‘new’ isolates and 1133 previously published isolates. They confirmed some well-established facts: parasites from each continent are genetically distinct, parasites from South America are related to Africa where they likely came from.

Interestingly, they confirm and expand a recent discovery from Otto et al that some var domains are conserved worldwide, and this is only partially explained by drug selective sweeps. This opens a new door of investigations as to what might drive this selection. The novelty of the study lies in the algorithm used to analyse similarity between DBLa types, taking into account recombinations. This improved and more biologically relevant analysis will likely be used by other labs in the future.

Although I wrote ‘major concerns’ here under, I think they are relatively straightforward to address and hopefully will improve this great piece of research.

Disclaimer: My expertise with JHMM, t-SNE, FFP and Viterbi algorithm is limited.

MAJOR

Fig1a explains very clearly why the previous method (PTS) can be problematic. Could you make another cartoon that describes the JHMM principle used in this study? Also, a flowchart would help to understand the pipeline used for the analysis.

About Fig 3. According to Larremore et al 2015, there are 94 DBLa domains from P reichenowi. If I understand correctly, only about ~1% of these domains match P.f. DBLa in specific countries. Considering the low number of P reichenowi DBLa sequences, is it relevant to compare them against each individual country (Lines 287 to 291)?

My understanding of the algorithm behind tSNE is limited, but I disagree with line 254, on presented figures (Fig S3 & S4), the clustering by country is not evident. On S3, half Thai isolates cluster within Uganda. Iran isolates are within the South America isolates (contrasting with Fig 2B where they were closer to Asia). On S4, Peru and PNG isolates cluster with Gabon.

I agree that, biologically, we expect geography to override the asympto/uncomplicated classification, however that is not apparent on the tSNE analysis.

Also, about Fig S3, Peru is on the uncomplicated malaria tSNE plot and should not be there.

Line 312, comparison with previous methods. To summarize crudely, the authors are comparing their JHMM + tSNE analysis to the previous PTS + phylogenetic tree / admixture. As the plots are different, it’s not intuitive to compare the previous and current methods. In your dataset, how many reads potentially show the scenario described in Fig1a? This would help me get a feeling for the necessity of the JHMM method. The Supplementary text is a bit confusing because the PCA is missing in Fig S5 and S6. Also, when Fig S9 is cited, it should be Fig S8. Finally, on the tSNE plot, have you labelled the Uganda samples with the 6 regions of origin? I presume there is no ‘intra-country’ cluster?

Line397. If I remember correctly the DBLa universal primers do not match type3 var genes. This would explain why they are not detected here. (Otto2019 found 680 copies of these genes in 714 isolates, so it would be very surprising if they don’t exist in the countries from this study)

Line 348-349: The most frequent DBLa types were selected as those present in > 50 isolates. However the number of isolates per country is very unevenly distributed. Does it mean that most of the highly frequent DBLa types are from Africa? Could you normalize the number of isolates per country (as in Otto 2019, for example)?

The authors claim that they can track DBLa types to specific localities, countries and continent, thus they can be used as markers of geographic patterns. In my humble opinion it still remains to be shown that these ‘DBLa signatures’ are stable over time. From one year to the next, due to high recombination within var genes, the DBLa signature could be quite different and making it impossible to track where an isolate originates (other than the continent level). It sounds like this is what Dr Ruybal-Pesantez is currently working on in Ecuador, and I’m very much looking forward to reading about it. But as of today I believe more temporal data is needed.

MINOR

- In general, for each supplementary figure, please make sure to indicate what dataset and what method was used in the figure legend. A workflow of the analysis would help.

- Line 100: I’m not sure if Ref20 is relevant here?

- The colour code (currently alphabetical) is not intuitive, it would be better to arrange it by geography.

- Fig1a, line 172. This might be a question of semantic, but to me the cartoon could result from a single recombination event (non-homologous translocation with reciprocal exchange)

- Line 207 and Fig 2A. I’m confused as why some isolates may have 19 or fewer DBLa because in the Methods (Line565) it says that isolates with fewer than 20 sequences are removed?

- Line 246, the paragraph investigates Multiplicity Of Infection as a confounding factor and mentions the number of DBLa types to infer the number of strains present in an isolate. Could you indicate that inferred MOI in Table 1? I’m surprised that the number of DBLa types is not associated with country of origin, we typically expect higher MOI in African countries than elsewhere.

- Line 382: Please rephrase ‘Every high-frequency type was seen in at least one Asian or African country, except for 19 types that were only seen in African countries.’

- Line410. What var gene is ‘NTSB3-DBLα0.4-CIDRα6-DBLβ5-DBLγ10-DBLδ6-CIDRβ2 ‘? Is it conserved in 3D7? If so what’s its ID?

- Line 495: Was the sample collection approved by a local Ethics Committee? (for the new sample collections)

- Line 536: Why using the term ‘read’ when talking about a translated sequence?

- Line 552: According to Table 1, Venezuela is the smallest dataset, not Colombia.

- Line 568: ‘… a threshold of 2 DBLa types’.

- Citation format is not consistent (e.g. line 103)

- Something went wrong with many of the references. Some include the word ‘ [Internet]’ and the URL.

- Fig 4. What is the difference between grey and black dots? Could you indicate UpsA / non-upsA groups?

- Fig S5. The title of that figure is the exact same as Fig 2C so it’s a bit confusing. I’m not sure if FigS5 is really necessary.

- Fig S6: the legend is confusing, please rephrase.

- Fig S7. In the legend, the description of the two plots is inverted. Is the recombination hotspot in the middle of the DBLa domain corresponding to the relatively conserved “REDWW” domain?

- Fig S14. X and Y axis labels, replace ‘gene’ by ‘DBLa type’.

- Fig S15. I presume the DBLa type order is the same in panel A and B? Please indicate. Actually, there are two Figures S15…

- Fig S16 named S15 and line 569 : ‘some clustering by coverage (x-axis) is evident’), I can’t see it on the plot.

- Table S1. Columns C and S have no header. SampleID missing in D7. Please indicate what the bold / blue /red colour scheme corresponds to.

What is “FCR3var3” in column C? Because in the main manuscript it says that type3 var were not part of the top100 DBLa types.

Did you use the 15 PacBio genome assemblies from Otto2018? This might help identify chromosomal location of some of the conserved DBLa.

- SuppDataFile1.xlsx needs some information about what each spreadsheet is.

- SuppDataFile3.txt, the header line is missing.

- Does file ‘https://raw.githubusercontent.com/gtonkinhill/global_var_manuscript/master/data/combined_all_454_noPOR.fasta’ contain all DBLa types used in this manuscript?

**Have all data underlying the figures and results presented in the manuscript been provided?**

Reviewer #1: Yes

Reviewer #2: Yes

PLOS authors have the option to publish the peer review history of their article (what does this mean?). If published, this will include your full peer review and any attached files.

Reviewer #1: No

Reviewer #2: **Yes: **Antoine Claessens

---

## [Decision Letter · Decision Letter 1]

10 Nov 2020

Dear Dr Day,

We are pleased to inform you that your manuscript entitled "Evolutionary analyses of the major variant surface antigen-encoding genes reveal population structure of  Plasmodium falciparum  within and between continents" has been editorially accepted for publication in PLOS Genetics. Congratulations!

Yours sincerely,

Carmen Buchrieser

Associate Editor

PLOS Genetics

Hua Tang

Section Editor: Natural Variation

PLOS Genetics

Comments from the reviewers (if applicable):

Reviewer's Responses to Questions

**Comments to the Authors:**

Reviewer #1: All my queries were very well responded or rebutalled. I like the new figue one and the comparsion with the laverania is very nice. The slightly higher similarity of the P. praefalciparum to Iran than Africa is intruiging but maybe within the expected error margen.

Reviewer #2: The Authors have addressed all my previous concerns. I thorouhly recommend publication.

**Have all data underlying the figures and results presented in the manuscript been provided?**

Reviewer #1: Yes

Reviewer #2: Yes

PLOS authors have the option to publish the peer review history of their article (what does this mean?). If published, this will include your full peer review and any attached files.

Reviewer #1: No

Reviewer #2: **Yes: **Antoine Claessens

**Data Deposition**

http://datadryad.org/submit?journalID=pgenetics&manu=PGENETICS-D-20-00892R1

**Press Queries**

---

## [Editor Report · Acceptance letter]

22 Jan 2021

PGENETICS-D-20-00892R1 

Evolutionary analyses of the major variant surface antigen-encoding genes reveal population structure of  Plasmodium falciparum  within and between continents 

Dear Dr Day, 

We are pleased to inform you that your manuscript entitled "Evolutionary analyses of the major variant surface antigen-encoding genes reveal population structure of  Plasmodium falciparum  within and between continents" has been formally accepted for publication in PLOS Genetics! Your manuscript is now with our production department and you will be notified of the publication date in due course.

With kind regards,

Alice Ellingham

PLOS Genetics

On behalf of:
